# Cryogenic strength improvement by utilizing room-temperature deformation twinning in a partially recrystallized VCrMnFeCoNi high-entropy alloy

Y.H. Jo[1], S. Jung[1], W.M. Choi[1], S.S. Sohn[1], H.S. Kim[1], B.J. Lee[1], N.J. Kim[2] & S. Lee[1]

The excellent cryogenic tensile properties of the CrMnFeCoNi alloy are generally caused by deformation twinning, which is difficult to achieve at room temperature because of insufficient stress for twinning. Here, we induced twinning at room temperature to improve the cryogenic tensile properties of the CrMnFeCoNi alloy. Considering grain size effects on the critical stress for twinning, twins were readily formed in the coarse microstructure by cold rolling without grain refinement by hot rolling. These twins were retained by partial recrystallization and played an important role in improving strength, allowing yield strengths approaching 1 GPa. The persistent elongation up to 46% as well as the tensile strength of 1.3 GPa are attributed to additional twinning in both recrystallized and non-recrystallization regions. Our results demonstrate that non-recrystallized grains, which are generally avoided in conventional alloys because of their deleterious effect on ductility, can be useful in achieving high-strength high-entropy alloys.

[1] Center for High Entropy Alloys, Pohang University of Science and Technology, Pohang 790-784, Korea. [2] Graduate Institute of Ferrous Technology, Pohang University of Science and Technology, Pohang 790-784, Korea. Correspondence and requests for materials should be addressed to S.S.S. (email: bbosil7@postech.ac.kr).

New unique alloys with five or more elements present in similar portions within the alloy that have preferentially solid-solution phases have been developed as a class of high-entropy alloys (HEAs)[1–4]. These HEAs exist as single multi-element solid solutions, have excellent thermal stabilities[5–7], are generally composed of a single phase of face centered cubic (fcc) or body centered cubic (bcc), and have properties that vary depending on the types and amounts of alloying elements[8–11]. Among these HEAs, an equi-atomic CrMnFeCoNi five-component alloy developed by Cantor et al.[12] features strongly temperature-dependent strength and ductility, and its cryogenic-temperature strength and ductility are much higher than those at room-temperature because of its fcc structure and deformation twinning[13–17]. HEAs show better fracture toughness and corrosion resistance at cryogenic temperatures than conventional stainless steels used for cryogenic applications[15–19]. Because the properties of HEAs, including the CrMnFeCoNi alloy, correspond well with those of structural materials for cryogenic extreme-environmental applications, the successful replacement of 9%-Ni and stainless steels with HEAs can be expected.

The excellent tensile properties of the CrMnFeCoNi alloy at cryogenic temperatures are mainly caused by deformation twinning[14,15,20]. However, such twinning is difficult to achieve at room temperature because the resolved shear stress is insufficient to reach the critical stress for twinning[20,21]. Another issue is the relatively low yield strength (0.2–0.6 GPa) caused by the fcc structural characteristics of the alloy[14,20,22–24]. Here, we effectively induce twinning in an HEA at room temperature and exploit the induced twinning to improve the cryogenic tensile properties, particularly the yield strength, of the material. Because the critical twinning stress depends on the grain size, which provides a significant potential for twinning at room temperature[25–27], the investigated HEA was cold rolled right after a homogenization treatment without a hot-rolling process and was annealed at 750 and 900 °C for 10 min to obtain partially and fully recrystallized microstructures, respectively. Due to the utilization of non-recrystallized grains containing retained deformation twins, the HEA showed ultra-high yield and tensile strengths of 0.97 and 1.3 GPa, respectively, together with a good ductility of 46%, which surpassed the tensile properties of the existing HEAs. Our results demonstrate that the existence of non-recrystallized regions, which are generally avoided in conventional alloys because of their deleterious effect on ductility, can provide a useful design concept in high-strength HEAs.

## Results

### Computational thermodynamic approach for alloy design.
The first objective of this research was to develop new VCrMnFeCoNi alloys for use in cryogenic applications. To achieve the chemical compositions of these alloys, a computational thermodynamic approach of a VCrMnFeCoNi six-component system was used, from which HEA compositions having a wide temperature range of an fcc single phase were obtained. This thermodynamic-approach-based alloy design was very effective in comparison with conventional alloying methods, that is, empirical and trial-and-error-based alloy design and property evaluation, which demand significant time and are expensive[28,29].

Vanadium was selected as a new candidate alloying element in the CrMnFeCoNi alloy system because it shows a wide single-phase solid-solution range on binary phase diagrams with most elements, including Cr, Fe and Mn. Although most HEAs have equi-atomic or near-equi-atomic compositions, the equi-atomic composition is not believed to be the optimum composition for a wide range of materials properties. The CrMnFeCoNi alloy system is not a unique system that yields fcc-single-phase microstructures containing more than five components[30–35]. The atomic sizes of all elements in the CrMnFeCoNi alloy are similar to each other, while the atomic size of vanadium is somewhat larger than the other elements. In addition, vanadium could have a larger negative mixing enthalpy and thus possess stronger bonds with Fe, Co and Ni than those among Cr, Mn, Fe, Co and Ni (ref. 36). The addition of vanadium could induce a larger solid-solution hardening effect.

The design of a new VCrMnFeCoNi alloy and the theoretical confirmation of an fcc-single-phase equilibrium were carried out with Thermo-Calc[37] software along with TCFE2000 and its upgraded version[38,39]. Figure 1a shows the equilibrium phase diagram obtained when mole fractions of Fe, Mn and Ni are varied in an alloy containing 10Co15Cr10V (at.%) in the temperature range of 600–800 °C. The compositions located near the boundary lines of 'Region 1' have a fcc single phase in the temperature range of 600 °C-melting point, while the compositions located in the right side of 'Region 1' have two phases (fcc + bcc). Based on this three-component phase diagram, the 35Fe5Mn25Ni (at.%) composition was selected to obtain a stable fcc single phase in the temperature range of 590 °C-melting point. Figure 1b shows the calculated mole fractions of the equilibrium phases in the temperature range of 300–1,400 °C. The present 10V15Cr5Mn35Fe10Co25Ni alloy has a very wide fcc-single-phase region (gray-colored area) in the temperature range of 590–1,300 °C.

### Microstructures.
X-ray diffraction patterns of the 750 °C- and 900 °C-annealed alloys (H750 and H900 alloys) as well as the as-homogenized alloy are shown in Fig. 1c. Peaks of an fcc single phase are only observed without any peaks of Co–Fe bcc or Cr-rich bcc phases, which indicates that the experimental fcc-single-phase microstructures are well matched with those estimated from thermodynamic calculations (Fig. 1a,b).

Figure 2a–c shows the electron backscatter diffraction (EBSD) inverse pole figure (IPF) maps of the as-homogenized, H900 and H750 alloys. The as-homogenized alloy consists of very coarse fcc-single-phase grains of ~157 μm in size (Fig. 2a). Equiaxed, recrystallized fcc-single-phase grains of ~5.2 μm in size (twin boundaries were not counted in the measurements) are homogeneously distributed in the H900 alloy (Fig. 2b). In the H750 alloy, fine recrystallized fcc grains (average grain size: ~1.5 μm) are mixed with coarse non-recrystallized fcc grains (average grain size: ~32 μm; Fig. 2c). Most of the fine recrystallized grains are linearly connected along the 35–65° direction deviating from the rolling direction because the recrystallization occurs preferentially along the 35–65° shear bands formed during cold rolling. The volume fraction of the recrystallized region is ~40%. Figure 2d,e shows higher-magnification EBSD IPF and image quality (IQ), with kernel average misorientation (KAM) maps of the partially recrystallized microstructure in the H750 alloy. Fine recrystallized grains are mostly aligned along the 35–65° direction, and several deformation twins having 60°-twin orientations within the matrix are observed in the non-recrystallized grains (Fig. 2d). These twins are formed during cold rolling and are aligned nearly along the rolling direction, which does not match the orientations of the recrystallized grains. This result suggests that deformation twins can be formed at room temperature, which is discussed in the present study. According to the IQ-KAM map (Fig. 2e), two types of grains are clearly distinguished. The KAM values of the recrystallized grains are lower than 1°, whereas those of the non-recrystallized grains are much higher (1–5°). The non-recrystallized grains are recovered, and the KAM values are somewhat higher near twinned areas than in matrix areas.

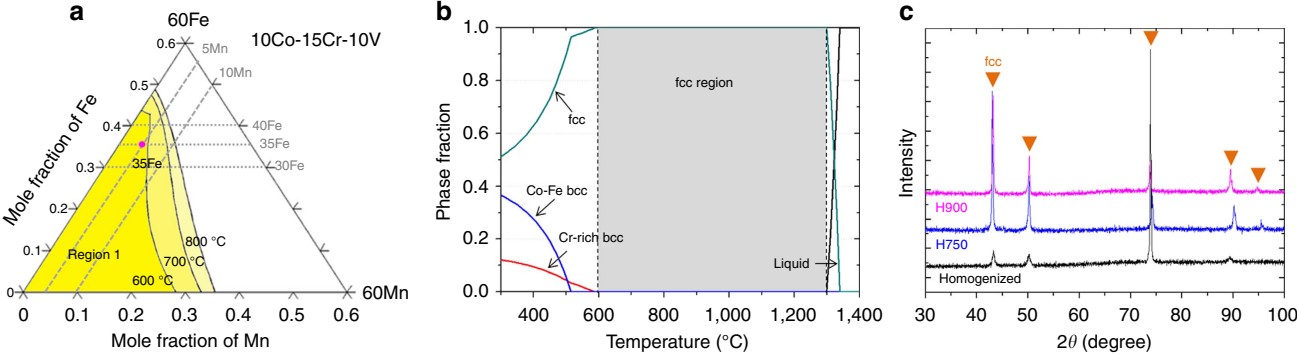

**Figure 1 | Equilibrium phase diagrams and experimental microstructure of the designed alloy.** (**a**) Equilibrium phase diagram obtained for varied mole fractions of Fe, Mn and Ni in an alloy containing 10Co15Cr10V in the temperature range of 600–800 °C. (**b**) Mole fractions of equilibrium phases of liquid, fcc, Co–Fe bcc and Cr-rich bcc in the temperature range of 300–1,400 °C. The 10V15Cr5Mn35Fe10Co25Ni alloy has a very wide fcc-single-phase region (gray-colored area) in the temperature range of 590–1,300 °C. (**c**) X-ray diffraction patterns of the 750 °C- and 900 °C-annealed alloys (H750 and H900 alloys). Peaks of an fcc single-phase are only observed without any peaks of Co–Fe bcc or Cr-rich bcc phases.

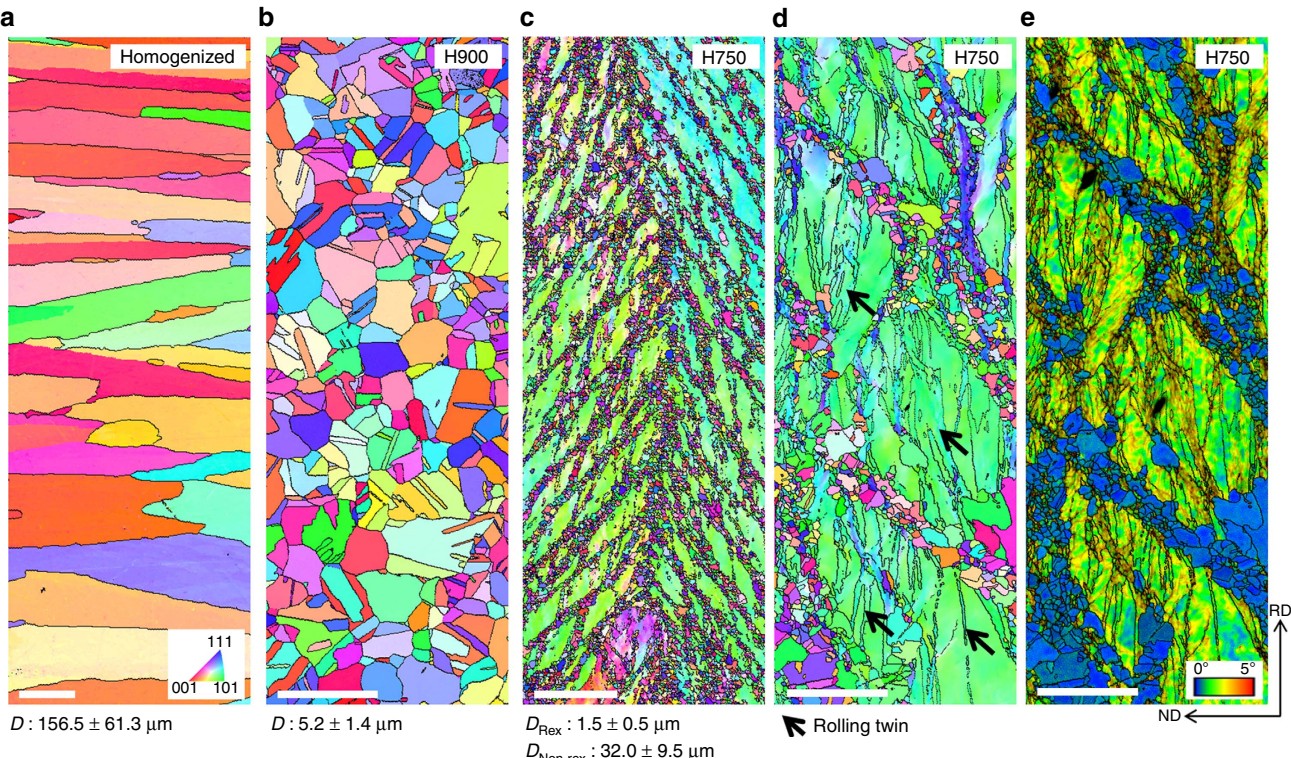

**Figure 2 | Microstructures of as-homogenized and annealed alloys.** (**a**) IPF map from an EBSD scan of the as-homogenized alloy, showing very coarse fcc-single-phase grains (average grain size: 157 μm). (**b**) IPF map of the H900 alloy, showing recrystallized fcc-single-phase grains (average grain size: 5.2 μm). (**c**) IPF map of the H750 alloy, showing fine recrystallized fcc grains (average grain size: ∼1.5 μm) mixed with coarse non-recrystallized fcc grains (average grain size: 32 μm). (**d,e**) Higher-magnification IPF and IQ with KAM maps of the partially recrystallized microstructure in the H750 alloy. The fine recrystallized grains are mostly aligned along the 35–65° direction, and a number of deformation twins with a 60°-twin orientation to the matrix are observed at the non-recrystallized grains. The scale bars in **a–e** are 100, 10, 60, 10 and 10 μm, respectively.

This difference indicates that the dislocation density is relatively high near twins retained after cold rolling, which can lead to improvements in yield and tensile strength.

**Tensile properties.** Figure 3 shows the room- and cryogenic-temperature engineering stress–strain curves of the H750 and H900 alloys. At room temperature, the H900 alloy shows a yield and tensile strengths of 498 and 752 MPa, respectively, along with an elongation of 52.5%. The strengths are higher in the H750 alloy than in the H900 alloy, while the elongation is lower (28%). At cryogenic temperatures, the strengths and elongation are improved over the room-temperature values. In the H900 alloy, the yield strength, tensile strength and elongation are 698 MPa, 1,128 MPa and 78.6%, respectively, at cryogenic temperatures. Higher yield and tensile strengths (970 and 1,314 MPa) and lower elongation (46.3%) are also shown in the H750 alloy at cryogenic temperature. Here, the yield strength of

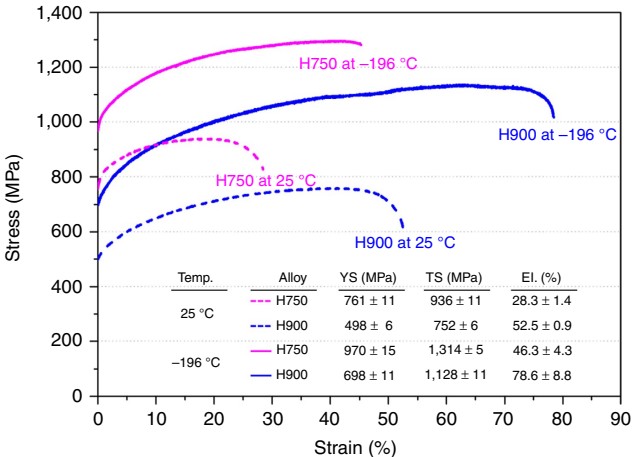

**Figure 3 | Room- and cryogenic-temperature tensile properties.**
Room- and cryogenic-temperature engineering stress–strain curves of the H750 and H900 alloys. At cryogenic temperatures, the strengths and elongation are improved over the room-temperature values. Higher yield and tensile strengths and lower elongation are shown in the H750 alloy. (The data points shown are the mean ± s.d. from 3 repeated measurements).

970 MPa is a notable property. Yield strengths on the order of 1 GPa have only rarely been achieved in existing annealed HEAs[13–15,20].

**Twin-related deformation mechanisms.** To explain the strength improvement as well as the excellent cryogenic tensile properties shown in the fully recrystallized H900 and partially recrystallized H750 alloys, detailed tensile deformation behaviour is essentially needed. Figure 4a–h shows EBSD IPF maps, IQ maps, transmission electron microscopy (TEM) bright- and dark-field (BF and DF) images, and selected-area diffraction (SAD) patterns of the cross-sectional area beneath the room- and cryogenic-temperature tensile-fractured surface of the H900 alloy. In the room-temperature IPF and IQ maps (Fig. 4a,b), deformation twins are hardly found. In the TEM BF image, this area consists of high-density tangled dislocations without any twins (Fig. 4c). This observation is confirmed from the SAD pattern taken along the [011] zone axis of an fcc grain (Fig. 4d), which shows no twinning spots. At cryogenic temperatures, deformation twins are clearly observed, as indicated by the arrows in Fig. 4e. When twins are thinly formed, their definition is quite poor in the IPF map. According to the IQ map (Fig. 4f), several sharp twins are observed. These twins are also found in the TEM DF image (Fig. 4g). Many parallel twins with thicknesses of 10–30 nm were

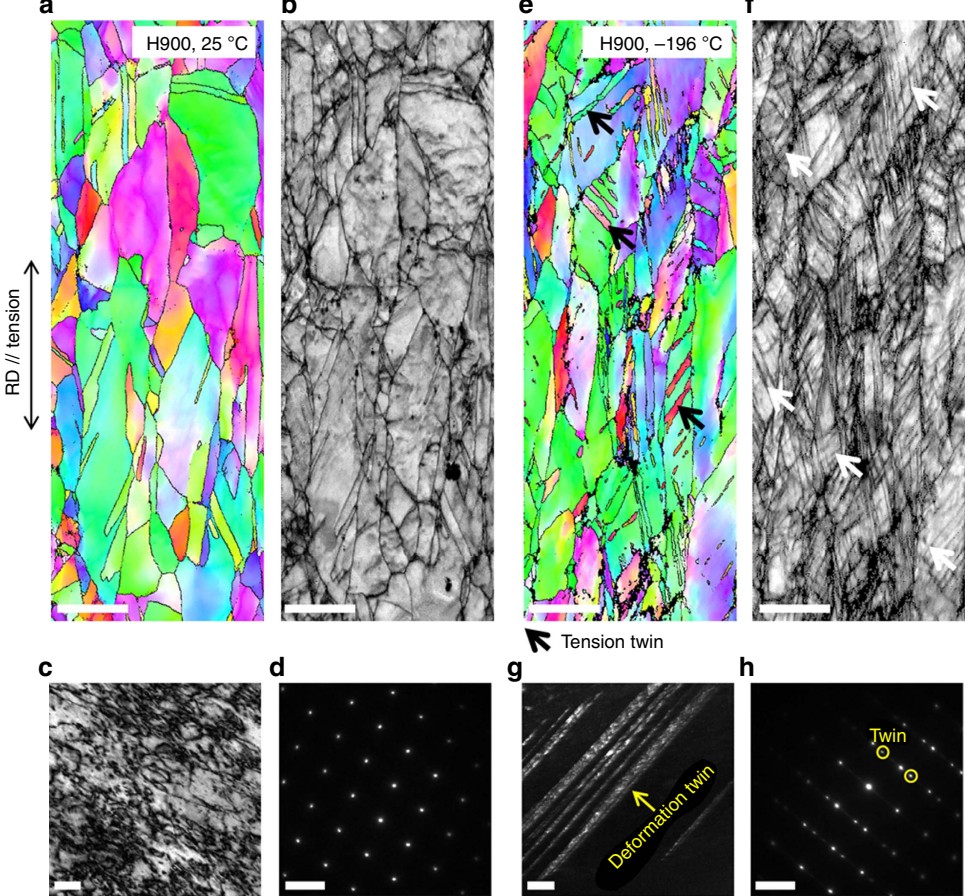

**Figure 4 | Room- and cryogenic-temperature tensioned microstructures of H900 alloy.** (**a–d**) IPF and IQ maps from an EBSD scan, TEM BF image and SAD pattern of the cross-sectional area beneath the room-temperature tensile-fractured surface showing very few deformation twins. The scale bars in **a–d** are 5 μm, 5 μm, 50 nm and 5 nm$^{-1}$, respectively. (**e–h**) EBSD IPF map, IQ map, TEM DF image and SAD pattern of the cross-sectional area beneath the cryogenic-temperature tensile-fractured surface, showing many deformation twins. The scale bars in **e–h** are 5 μm, 5 μm, 50 nm and 5 nm$^{-1}$, respectively.

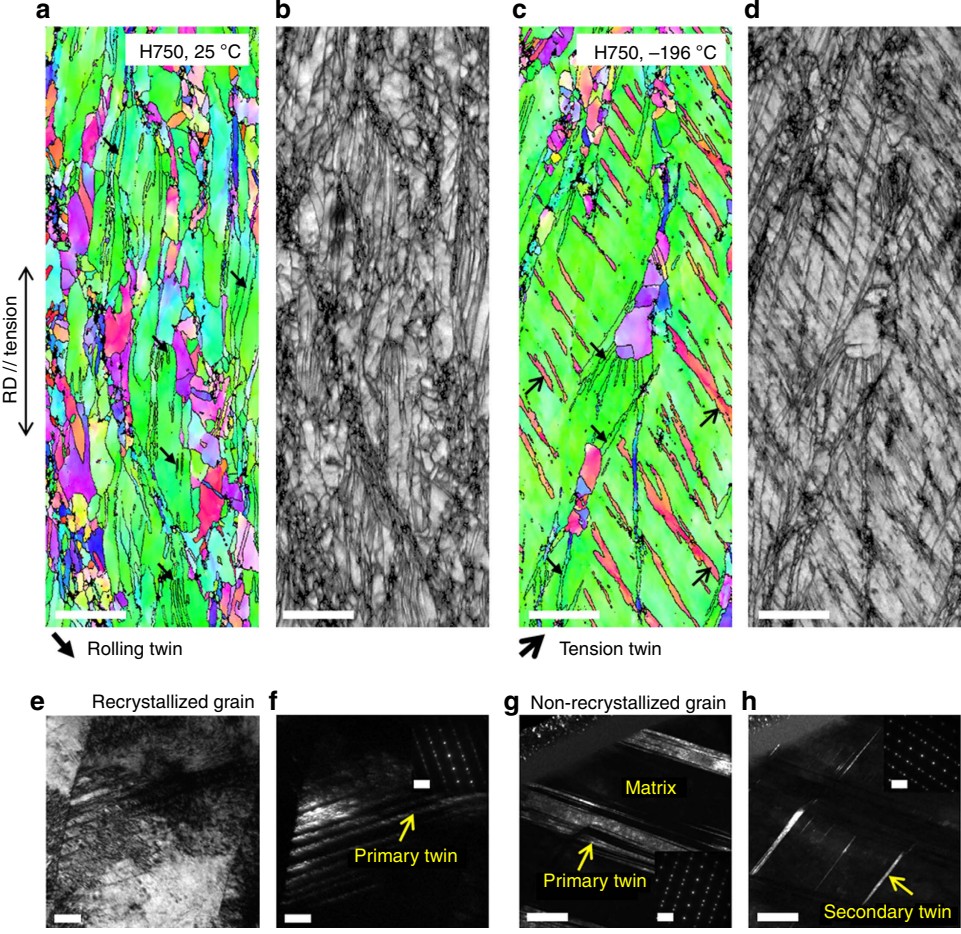

**Figure 5 | Room- and cryogenic-temperature tensioned microstructures of H750 alloy.** (**a,b**) IPF and IQ maps from an EBSD scan of the cross-sectional area beneath the room-temperature tensile-fractured surface, showing that most twins are retained after cold rolling, which indicates that there are no or few additional twins formed. The scale bars in **a,b** are 5 μm. (**c–h**) IPF map, IQ map, TEM BF and DF images, and SAD patterns of the cross-sectional area beneath the cryogenic-temperature tensile-fractured surface, showing many primary deformation twins at the recrystallized grain and a further-increased number of primary and secondary deformation twins at the non-recrystallized grain. The scale bars in **c–h** are 5 μm, 5 μm, 50 nm, 50 nm, 500 nm and 500 nm, respectively. The scale bars in the insets of **f,h** are 5 nm$^{-1}$.

observed, which was confirmed by the twinning spots in the SAD pattern (Fig. 4h); however, these twins did not develop into secondary twins.

Figure 5a–d shows the EBSD IPF and IQ maps of the cross-sectional area beneath the room- and cryogenic-temperature tensile-fractured surface of the H750 alloy. As shown in Fig. 2d,e, the H750 alloy shows a partially recrystallized microstructure mixed with fine recrystallized grains and coarse non-recrystallized grains before the tension. A considerable number of twins formed during cold rolling are retained inside the non-recrystallized grains. After the tensile deformation at room temperature, many twins are observed (Fig. 5a,b), but most are twins retained after cold rolling, which indicates that there are no or few additional twins formed at room temperature. At cryogenic temperatures, a number of parallel twins having 60°-twin orientation are observed at the non-recrystallized grains, and the number and area fraction of twins are larger than those before the tension (Fig. 5c,d), which indicates that new twins having different orientations from the twins retained after cold rolling are formed, as marked by the arrows. Figure 5e–h shows BF and DF images and SAD patterns of the recrystallized and non-recrystallized grains of the cryogenic-temperature tensioned H750 alloy. Fine twins of 10–30 nm in thickness are

formed in one primary system at the recrystallized grain (Fig. 5e,f), similarly to the cryogenic-temperature tensioned H900 alloy (Fig. 4g,h). At the non-recrystallized grain, however, primary twin bundles of ∼500 nm in thickness (Fig. 5g) are formed together with fine secondary twins of 30–70 nm in thickness (Fig. 5h). These TEM observations confirm that additional twins are formed after the cryogenic-temperature tension at the non-recrystallized grains.

## Discussion
The present 10V15Cr5Mn35Fe10Co25Ni alloy with a non-equiatomic composition is newly designed by thermodynamic calculations. An fcc single-phase microstructure is achieved after homogenization and annealing (Fig. 1c), which is in good agreement with the thermodynamic calculation results (Fig. 1b). When the grains are fully recrystallized (average grain size: ∼5.2 μm), the major deformation mechanism is a dislocation slip at room temperature, which changes to deformation twinning at cryogenic temperatures, similarly to the CrMnFeCoNi alloy. The cryogenic tensile properties (tensile strength: 1,128 MPa; elongation: 78.6%) are similar to the previously published data of the CrMnFeCoNi alloy[14]. Notably, the yield strength is

much higher (698 MPa) than the reported yield strengths (200–600 MPa) of the CrMnFeCoNi alloy with similar grain sizes. This difference might be because the addition of vanadium plays a role in solid-solution hardening, which is the purpose of the present alloy design; however, other alloying elements are also varied. In the H750 alloy with a microstructure of mixed recrystallized and non-recrystallized grains, the effect of the non-recrystallized grains on yield strength improvement is quite obvious. Approximately 60 vol.% of the coarse-grained non-recrystallized region as well as the fine-grained recrystallized region raise the room- and cryogenic-temperature yield strengths by ~270 MPa over those of the H900 alloy or by ~370 MPa over those of the CrMnFeCoNi alloy[14].

The role of this non-recrystallized region in significantly improving the yield strength comes from the deformation twins formed during cold rolling. Notably, several thick twins are formed after cold rolling, as confirmed in the microstructure of the H750 alloy (Fig. 2d,e), whereas there are no deformation twins when the H900 alloy composed of recrystallized grains is tensioned at room temperature (Fig. 4a–d). This difference is attributed to the effect of grain size, which influences the critical stress for twinning. As indicated by Meyers et al.[26], a highly unique characteristic of twinning is that the critical stress for twinning ($\sigma_T$) is more dependent on the grain size ($d$) than the critical stress for slip ($\sigma_S$). For most cases, a Hall-Petch relationship is obeyed, where the slope for twinning ($k_T$) is higher than the slope for slip ($k_S$) in the following Hall-Petch equations:

$$\sigma_T = \sigma_{T0} + k_T d^{-1/2} \qquad (1)$$

$$\sigma_S = \sigma_{S0} + k_S d^{-1/2} \qquad (2)$$

As the grain size increases, $\sigma_T$ becomes smaller than $\sigma_S$, which provides a good explanation for the profuse twinning at large grains.

The recent results from El-Danaf et al.[40] reconfirm the significant effect of grain size on the propensity for twinning. A 70/30 brass with an average grain size of 250 μm showed a much higher twinning density than that with average grain sizes of 9 or 30 μm. A 35Ni35Co20Cr10Mo (at.%) alloy with an average grain size of 40 μm readily showed twinning, whereas the same alloy with an average grain size of 1 μm did not show any evidence of twinning[40]. Meyers et al.[41] performed shock-compression experiments on copper and obtained profuse twins when its average grain sizes were 117 or 315 μm, while no twins were found for an average grain size of 9 μm. These results indicate that grain coarsening promotes twinning rather than slip. Thus, twins are readily formed by cold rolling in the coarse microstructure without the grain refinement by hot rolling, similarly to the H750 alloy. However, the H900 alloy composed of fine recrystallized grains hardly obtains any twins.

The most important result for the H750 alloy is the presence of deformation twins inside the coarse non-recrystallized grains retained after cold rolling. Because these twins have a higher thermal stability than dislocations after the initial recovery occurring during annealing at 750 °C, these twins are retained at room temperature inside the coarse non-recrystallized grains, while the dislocation density is lowered by the formation of sub-structures[42–44]. Consequently, when considering the higher dislocation density near twins (Fig. 2e), these twins enhance the yield strength because they act as strong barriers against mobile dislocations. When the H750 alloy is tensioned at cryogenic temperatures, many new deformation twins are formed (tension twins), and twins are retained at the non-recrystallized grains at room temperature after cold rolling (rolling twins).

Because the system of newly formed cryogenic-temperature twins is different from that of room-temperature twins (Fig. 5a,c), the fraction of twins that can be formed during room-temperature deformation is likely saturated; however, the deformation mode during cold rolling is different from that during the tensile test[45]. Thus, the populated formation of cryogenic-temperature twins, together with deformation twins retained at room temperature, readily leads to a high strain-hardening effect and ultra-high yield and tensile strengths (970 and 1,314 MPa) at cryogenic temperatures (Fig. 3).

This study of deformation twins provides a good method for improving the cryogenic-temperature tensile properties in the present thermodynamically designed HEA. Considering the effects of grain size on the critical stress for twinning, twins are formed at room temperature by cold rolling right after homogenization without hot rolling, which can usually provide grain refinement. These twins are retained by partial recrystallization and utilized as a powerful mechanism for improving the cryogenic-temperature yield and tensile strengths. The tensile properties of the present annealed HEAs exceed, to our knowledge, properties reported so far to date in past studies on HEAs. In particular, the present ultra-high yield strength (near-1 GPa grade) is achieved mainly by the existence of non-recrystallized grains, and the yield strength also benefits from the formation of fine recrystallized grains. The persistent elongation of up to 46% as well as the ultra-high yield and tensile strengths in the H750 alloy are attributed to the additional twinning in both the recrystallized and non-recrystallization regions. The merits of a good combination of strength and ductility and a simple manufacturing process of the present 10V15Cr5Mn35Fe10Co25Ni alloy are promising as new applications in ultra-high-strength HEAs, particularly those at cryogenic temperatures. Our results also reveal that room-temperature deformation twinning coupled with non-recrystallization can offer a strengthening mechanism for ultra-high-strength HEAs.

## Methods

**Fabrication of 10V15Cr5Mn35Fe10Co25Ni HEA.** An alloy designed by thermodynamic calculations was fabricated by compact vacuum induction melting equipment (model: MC100V, Indutherm, Walzbachtal-Wossingen, Germany) under an argon atmosphere. The master alloys were prepared from commercially pure elements (the purity of each raw material was at least 99.9%). The raw elements were alloyed in a ZrO$_2$-coated Al$_2$O$_3$ ceramic crucible in a vacuum induction melting furnace. The Al$_2$O$_3$ crucible was heated to 600 °C for 1 h to remove the water vapour before placing it into a furnace. The pouring temperature was set to 1,500 °C. Approximately 150 g of the master alloy was melted, superheated and poured into a rectangular graphite module with a length of 100 mm, width of 35 mm and thickness of 8 mm. Before melting, the furnace chamber was evacuated to $6 \times 10^{-2}$ Pa and backfilled with high-purity argon gas to reach 0.06 MPa. Each alloy ingot was melted three or four times to ensure compositional homogeneity. The ingots were homogenized at 1,100 °C for 6 h, pickled in 20% HCl and milled to a thicknesses of 7 mm. The pickled ingots were rolled at room temperature (reduction ratio: 75%) to produce 1.5-mm-thick sheets. The alloy sheets were annealed at 750 and 900 °C for 10 min to obtain partially and fully recrystallized microstructures, respectively, and were quenched with water.

**Microstructure characterization.** The phases present in the alloy sheets were identified by X-ray diffraction (Cu K$_\alpha$ radiation, scan rate: 2 deg per min, scan step size: 0.02 deg). EBSD analysis (step size: 0.07 μm) was also conducted using a field emission scanning electron microscope (FE-SEM, Quanta 3D FEG, FEI Company, USA). EBSD specimens were mechanically polished and electro-polished at room temperature in a solution of CH$_3$COOH (92%) and HClO$_4$ (8%) at an operating voltage of 32 V. The data were interpreted by orientation imaging microscopy analysis software provided by TexSEM Laboratories, Inc. The KAM was calculated up to the fifth neighbour shell with a maximum misorientation angle of 5° (ref. 46). The KAM maps served as a measure of the deformation-induced local orientation gradients inside the grains. Deformed microstructures were identified by TEM (model: 2,100, JEOL, Japan) at an acceleration voltage of 200 kV.

Focused-ion beam (FIB, model: Quanta 3D FEG, FEI Company, USA) was used to prepare TEM thin foils.

**Mechanical property tests.** Plate-type sub-sized tensile specimens with a gauge length of 6.4 mm, gauge width of 2.5 mm and gauge thickness of 1.5 mm were prepared in the longitudinal direction. The specimens were tested at room and cryogenic temperatures ($-196\,^{\circ}\mathrm{C}$) at a crosshead speed of $6.4 \times 10^{-3}\,\mathrm{mm\,s^{-1}}$ by using a universal testing machine (model: 8,801, Instron, Canton, MA, USA) with a 100-kN capacity. A low-temperature chamber (size: $50 \times 40 \times 38$ cm) was attached into the universal testing machine in the case of the cryogenic-temperature tensile test by placing the specimen in liquid $N_2$ and equilibrating it. The representative data were obtained by averaging three values at each datum point and are reported with the s.d.

**Data availability**. The data that support the findings of this study are available from the corresponding author upon reasonable request.

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

# Acknowledgements

We thank Dr A. Zargaran of POSTECH for the TEM analysis. This work was supported by the Future Material Discovery Project of the National Research Foundation of Korea (NRF) funded by the Ministry of Science, ICT and the Future Planning (MSIP) of Korea (NRF-2016M3D1A1023383) and by the Brain Korea 21 PLUS Project for Center for Creative Industrial Materials.

## Author contributions

Y.H.J., S.S.S., H.S.K., N.J.K. and S.L. designed the experiments. Y.H.J. and S.J. performed all the experiments. W.M.C. and B.J.L. conducted the thermodynamic calculations. Y.H.J., S.S.S., N.J.K. and S.L. analysed the data. S.S.S., B.J.L., N.J.K. and S.L. wrote the manuscript with contributions from the other authors. All authors commented on the final manuscript and conclusions of this work.

## Additional information

**Competing interests:** The authors declare no competing financial interests.

