## [Peer Review File · Nature Communications]

Reviewers' Comments:

Reviewer #1 (Remarks to the Author):

The paper reported a V containing high entropy alloy, which was cold rolled and then annealing, which make the high entropy alloy has improved yield strength at room temperature and low temperature, the properties are better than the Cantor high entropy alloys. The interaction between the deformation twins and rolling twins, which make the materials wonderful, the paper is very interesting, however, there are some minor revisions needed:

[1] In Figure 3, H900 at 196, should be -196; H900 at -25, should be 25;

[2] Two papers related to the low temperature properties of high entropy alloys should be cited:

2.1 Title: High-entropy Al_{0.3}CoCrFeNi alloy fibers with high tensile strength and ductility at ambient and cryogenic temperatures

Author(s): Li, Dongyue; Li, Chengxin; Feng, Tao; et al.

Source: Acta Materialia Volume: 123 Pages: 285-294 Published: JAN 15 2017

2.2 Title: The ultrahigh charpy impact toughness of forged Al_xCoCrFeNi high entropy alloys at room and cryogenic temperatures

Author(s): Li, DongYue; Zhang, Yong

Source: Intermetallics Volume: 70 Pages: 24-28 Published: MAR 2016

Reviewer #2 (Remarks to the Author):

I do not recommend this paper, because the concept by partially recrystallized microstructures to improve the strength is well adapted in the material engineering field. This is just similar to heavily cold work imposed and the ductility is recovered to some degree by recovering/recrystallization. It is clear in the manuscript that the ductility by partial recrystallized will be much less than the fully recrystallized material. This is just an example to achieve the strength by compromise the ductility.

Therefore, the paper should be considered some material engineering journal, such as, Materials science and engineering A.

Reviewer #3 (Remarks to the Author):

The research is novel by several reasons:

1. Use a new alloy design route based on 10V-15Cr-5Mn to find 10V15Cr5Mn35Fe10Co25Ni which is non-equiatomic 6-component HEA but has a thermal stable FCC phase in a wide temperature range 590~1300°C.
2. Use a new route to get partially recrystallized structure with retained twins formed by cold rolling as-homogenized coarse-grained sample. This lets the alloy exhibit superior mechanical properties at -196 °C as compared with those reported for FCC alloys based on the Co-Cr-Fe-Mn-Ni system.
3. Fully discuss the reason why coarse-grained structure can enhance the formation of twins during cold rolling, by which they produce partially recrystallized structure by 750°C-annealing and recrystallized structure by 900°C-annealing.
4. Adequately discuss the reason why 750°C-annealed alloy (H750) have the outstanding mechanical properties. EBSD images and TEM analyses also give persuasive supports.

Therefore, the manuscript is strongly recommended to be published in Nature communications.

However, several corrections are suggested:

1. In Fig. 3, check the errors about "at 25°C" and "at -196°C"
2. Specific composition expression 10V15Cr5Mn35Fe10Co25Ni is better. Hyphen "-" could be deleted for clarity.
3. In line 16, over 1 GPa should be near 1 GPa.
4. In lines 25-27, it is better to use "New unique alloys preferentially having solid solution phases have been developed as a name of high entropy alloy (HEA) when five or more elements are alloyed with a similar portion of each element¹⁻⁴. These HEAs might have a single multi-element solid solution and show excellent thermal stability⁵⁻⁷."
4. In line 47, 0.96 GPa should be 0.97 GPa.
5. In line 67, "... than the other element. In addition, vanadium could have larger negative mixing enthalpies and thus stronger bondings with Fe, Co, and Ni than those among Cr, Mn, Fe, Co and Ni [ref: cohesion in metals]. A larger solid solution hardening effect....." is more persuasive.
6. In line 166, improve the wording "About of 60 vol.% of the coarse grained non-recrystallized region.....".

Reviewers' comments:

Reviewer #1:

The paper reported a V containing high entropy alloy, which was cold rolled and then annealing, which make the high entropy alloy has improved yield strength at room temperature and low temperature, the properties are better than the Cantor high entropy alloys. The interaction between the deformation twins and rolling twins, which make the materials wonderful, the paper is very interesting, however, there are some minor revisions needed:

Q1> In Figure 3, H900 at 196, should be -196; H900 at -25, should be 25;

→ We agree with the reviewer. We have corrected the typos.

Q2> Two papers related to the low temperature properties of high entropy alloys should be cited:

2.1 Title: High-entropy Al_{0.3}CoCrFeNi alloy fibers with high tensile strength and ductility at ambient and cryogenic temperatures

Author(s): Li, Dongyue; Li, Chengxin; Feng, Tao; et al.

Source: Acta Materialia Volume: 123 Pages: 285-294 Published: JAN 15 2017

2.2 Title: The ultrahigh charpy impact toughness of forged Al_xCoCrFeNi high entropy alloys at room and cryogenic temperatures

Author(s): Li, DongYue; Zhang, Yong

Source: Intermetallics Volume: 70 Pages: 24-28 Published: MAR 2016

→ We have added the suggested papers (Refs. #16,17) in the “Introduction” part.

16. Dongyue, L. *et al.* High-entropy Al_{0.3}CoCrFeNi alloy fibers with high tensile strength and ductility at ambient and cryogenic temperatures. *Acta Mater.* **123**, 285-294 (2017).

17. Dongyue, L. & Yong, Z. The ultrahigh charpy impact toughness of forged Al_xCoCrFeNi high entropy alloys at room and cryogenic temperatures. *Intermetallics* **70**, 24-28 (2016).

Reviewer #2:

I do not recommend this paper, because the concept by partially recrystallized microstructures to improve the strength is well adapted in the material engineering field. This is just similar to heavily cold work imposed and the ductility is recovered to some degree by recovering/recrystallization. It is clear in the manuscript that the ductility by partial recrystallized will be much less than the fully recrystallized material. This is just an example to achieve the strength by compromise the ductility.

Therefore, the paper should be considered some material engineering journal, such as, Materials science and engineering A.

→ We agree with the reviewer because the present work shows an improvement in ductility by recrystallization (as compared to a partially recrystallized condition), which is similar to conventional alloys subjected to heavy cold work followed by recovery/recrystallization (*i.e.*, annealing). However, there is an important difference between the two cases. In the case of conventional alloys subjected to heavy cold working followed by annealing, the microstructures of the partially recrystallized alloys are much different from those of the cold-worked alloys because tangled dislocations present in the as-cold-rolled condition are rearranged to form recovered cell structures. There are some examples of partially recrystallized microstructures for austenitic TWIP steels [Ref. a], IN706 nickel alloys [Ref. b], and austenitic stainless steels [Ref. c], as shown below. In the present case, on the other hand, the twinned structures present in the as-cold-rolled condition are retained after the partial recrystallization so that many pre-existing twins can be favorably utilized. This gives an opportunity for the development of high entropy alloys (HEAs) having high strengths as well as good ductility at both room and cryogenic temperatures, which have been a major drawback of previously developed HEAs.

Ref a

Ref b

Ref c

Ref. a : Kang, S., Jung, Y.S., Jun, J.H. & Lee, Y.K. Effects of recrystallization annealing temperature on carbide precipitation, microstructure, and mechanical properties in Fe-18Mn-0.6C-1.5Al TWIP steel. *Mater. Sci. Eng. A* **527**, 745-751 (2010).

Ref. b : Shuo, H., Lei, W., Xintong, L., Beijiang Z. & Guangpu, Z. Development of Constitutive Equation and Processing Maps for IN706 Alloy. *Acta Metall. Sinica* **27**, 198-204 (2014).

Ref. c : Souza, R.C., Silva, E.S., Jorge Jr. A.M., Cabrera, J.M. & Balancin, O. Dynamic recovery and dynamic recrystallization competition on a Nb- and N-bearing austenitic stainless steel biomaterial: Influence of strain rate and temperature. *Mater. Sci. Eng. A* **582**, 96-107 (2013).

Reviewer #3:

The research is novel by several reasons:

1. Use a new alloy design route based on 10V-15Cr-5Mn to find 10V15Cr5Mn35Fe10Co25Ni which is non-equiatomic 6-component HEA but has a thermal stable FCC phase in a wide temperature range 590~1300°C.
2. Use a new route to get partially recrystallized structure with retained twins formed by cold rolling as-homogenized coarse-grained sample. This lets the alloy exhibit superior mechanical properties at -196 °C as compared with those reported for FCC alloys based on the Co-Cr-Fe-Mn-Ni system.
3. Fully discuss the reason why coarse-grained structure can enhance the formation of twins during cold rolling, by which they produce partially recrystallized structure by 750°C-annealing and recrystallized structure by 900°C-annealing.
4. Adequately discuss the reason why 750°C-annealed alloy (H750) have the outstanding mechanical properties. EBSD images and TEM analyses also give persuasive supports.

Therefore, the manuscript is strongly recommended to be published in Nature communications.

However, several corrections are suggested:

Q1> In Fig. 3, check the errors about "at 25°C" and "at -196°C"

→ We agree with the reviewer. We have corrected the typos.

Q2> Specific composition expression 10V15Cr5Mn35Fe10Co25Ni is better. Hyphen "-" could be deleted for clarity.

→ We agree with the reviewer. We have modified the expressions including 10V-15Cr-5Mn-35Fe-10Co-25Ni, Cr-Mn-Fe-Co-Ni, V-Cr-Mn-Fe-Co-Ni, 10V-15Cr-5Mn, 35Fe-5Mn-25Ni, and 35Ni-35Co-20Cr-10Mo.

Q3> In line 16, over 1 GPa should be near 1 GPa.

→ We agree with the reviewer. We have corrected it.

Q4> In lines 25-27, it is better to use "New unique alloys preferentially having solid solution phases have been developed as a name of high entropy alloy (HEA) when five or more elements are alloyed with a similar portion of each element¹⁻⁴. These HEAs might have a single multi-element solid solution and show excellent thermal stability⁵⁻⁷."

→ We agree with the reviewer. We have modified the sentences as the reviewer recommended.

Q5> In line 47, 0.96 GPa should be 0.97 GPa.

→ We agree with the reviewer. We have corrected it.

Q6> In line 67, "... than the other element. In addition, vanadium could have larger negative mixing enthalpies and thus stronger bondings with Fe, Co, and Ni than those among Cr, Mn, Fe, Co and Ni [ref: cohesion in metals]. A larger solid solution hardening effect....." is more persuasive.

→ We agree with the reviewer. We have added the sentences as the reviewer recommended. The following reference has also been added.

36. de Boer, F.R., Boom, R., Mattens, W.C.M., Miedema, A.R. & Niessen, A.K. *Cohesion in metals: Transition metal alloys* (North-Holland, Amsterdam, Netherlands, 1988).

Q7> In line 166, improve the wording "About of 60 vol.% of the coarse grained non-recrystallized region.....".

→ We agree with the reviewer. We have modified 'coarse-grained' to 'coarse grained' as the reviewer recommended.

Figures and figure legends have also been revised according to the formatting guide for Nature Communications. We thank the reviewers for helpful comments, and feel certain that the paper has been improved by incorporating all the comments.

Sincerely yours,

Seok Su Sohn

Professor, POSTECH